# Pharmacognostic Study, Diuretic Activity and Acute Oral Toxicity of the Leaves of *Xiphidium caeruleum* Aubl. Collected in Two Different Phenological Stages

**DOI:** 10.3390/plants12061268

**Published:** 2023-03-10

**Authors:** Yamilet I. Gutiérrez Gaitén, Alejandro Felipe González, Ramón Scull Lizama, Carlos R. Núñez Cairo, Greisa L. Álvarez Hernández, Venancio Díaz Masó, Ana C. Noa Rodríguez, José A. Herrera Isidrón, Luc Pieters, Kenn Foubert, Lisset Herrera Isidrón

**Affiliations:** 1Instituto de Farmacia y Alimentos, Universidad de la Habana, La Habana 17100, Cuba; 2Instituto de Ciencias Básicas y Preclínicas “Victoria de Girón”, Universidad de Ciencias Médicas, La Habana 11300, Cuba; 3Laboratorios Farmacéuticos NOVATEC, La Habana 11300, Cuba; 4Hospital “Joaquín Albarrán Domínguez”, La Habana 10400, Cuba; 5Centro Nacional de Toxicología, La Habana 11300, Cuba; 6Instituto de Ciencia y Tecnología de Materiales, Universidad de la Habana, La Habana 10400, Cuba; 7Natural Products and Food Research and Analysis (NatuRA), University of Antwerp, 2610 Antwerp, Belgium; 8Unidad Profesional Interdisciplinaria de Ingeniería Campus Guanajuato (UPIIG), Instituto Politécnico Nacional, Av. Mineral de Valenciana 200, Puerto Interior, Silao de la Victoria 36275, Guanajuato, Mexico

**Keywords:** *Xiphidium caeruleum*, phytochemical composition, diuretic activity

## Abstract

*Xiphidium caeruleum* Aubl. is traditionally used in Cuba as an analgesic, anti-inflammatory, antilithiatic and diuretic remedy. Here we studied the pharmacognostic parameters of the leaves of *X. caeruleum*, the preliminary phytochemical composition, diuretic activity and acute oral toxicity of the aqueous extracts from the leaves of plants collected in the vegetative (VE) and flowering (FE) stages. The morphological characteristics and physicochemical parameters of leaves and extracts were determined. The phytochemical composition was assessed by phytochemical screening, TLC, UV, IR and HPLC/DAD profiles. The diuretic activity was evaluated in Wistar rats and compared to furosemide, hydrochlorothiazide and spironolactone. Epidermal cells, stomata and crystals were observed on the leaf surface. Phenolic compounds were identified as the main metabolites, including phenolic acids (gallic, caffeic, ferulic and cinnamic acids) and flavonoids (catechin, kaempferol-3-*O*-glucoside and quercetin). VE and FE showed diuretic activity. The activity of VE was similar to furosemide, and the activity of FE resembled spironolactone. No acute oral toxicity was observed. The presence of flavonoids and phenols in VE and FE may explain at least in part the traditional use and provide some insight into the reported ethnomedical use as a diuretic. Because of the differences in polyphenol profiles between VE and FE, further studies should be carried out to standardize the harvesting and extraction conditions in order to use *X. caeruleum* leaf extract as herbal medicine.

## 1. Introduction

Modern drug discovery is usually focused on pure chemical entities of synthetic origin, neglecting the use of whole plant extracts. According to the World Health Organization (WHO), 80% of the world’s population uses plants as a remedy to cure diseases [1], and around 25% of the existing drugs are derived from natural sources [2]. Whole plant extracts are used in traditional medicine, and in some particular cases, they have been acknowledged as producing better therapeutic results and fewer side effects than single pure compounds [3,4]. This observation has been explained by the additive or synergistic effects of many plant metabolites. However, scientific studies are necessary to corroborate the ethnobotanical information [5]. In addition, the search for new drug candidates from natural sources is often difficult because of the complex composition of plant extracts [3].

In Cuba, the great variety of medicinal plants, together with the wide ethnobotanical knowledge, create many opportunities for the search for new therapies in the treatment of various diseases. However, the phytotherapeutic potential of the island is still largely unexplored [6]. *Xiphidium caeruleum* Aubl. (Haemodoraceae) is a medicinal plant widely distributed in the Neotropics, ranging from Mexico to Northern South America, including the Antilles [7]. Leaves from this plant have been used in topical and internal formulations against snakebites [8,9], steam infusions are reported to facilitate childbirth and young leaf stalks are used for treating anemia, amenorrhea and other female infirmities [10,11]. *X. caeruleum* must not be used by pregnant women because of its abortive properties, but its use is encouraged in the postpartum diet. This is related to its use for treating amenorrhea and promoting the expulsion of ‘dead blood’ after delivery, which is due to the contraction of the uterus [11]. No toxic effects have been reported so far for this plant, and it is considered ‘not harmful’ to new mothers. In addition, it can help in the treatment of side effects caused by the consumption of other plant species [12,13]. In Cuba, *X. caeruleum* is used as an analgesic and anti-inflammatory remedy and for treating skin and renal diseases; the latter is ranked among the ten diseases with the highest mortality nationwide [14]. Nevertheless, neither pharmacological nor toxicological studies support its traditional use. Diuretics are among the most commonly used drugs for nephritic syndrome, cirrhosis, renal failure and hypertension toxemia during pregnancy. However, many synthetic diuretics have side effects such as fatigue, gout, ototoxicity, hyperglycemia and impotence. These factors favor the search for alternative, less toxic treatments [15].

*X. caeruleum* is chemically characterized by the presence of phenylphenalenones, which are considered to be chemical markers for the Haemodoraceae family [16]. Phenylbenzoisochromene, phenylbenzoisoquinoline and phenylcarbamoylnaphthoquinone have been reported from whole plant material. The presence of apigenin and related structures has been reported in leaves and flowers [17,18], while two phenylbenzoisolquinolindione alkaloids were isolated from the flowers [19]. Qualitative and quantitative variations in the phenylphenalenone composition in different plant parts have also been reported [20,21,22].

The life cycles of plants are divided into different phenological stages of development (germinative, vegetative, flowering, etc.) [23]. The present research was carried out to compare the pharmacognostic parameters of the leaves and to investigate the phytochemical composition, diuretic activity and acute oral toxicity of the aqueous extract of *X. caeruleum* leaves, collected in the vegetative and flowering stages. This is the first biological study on this species, and it provides the first evidence for pharmacognostic and pharmacological differences between these two phenological stages.

## 2. Results

### 2.1. Morphological Characteristics of X. caeruleum Leaves

The leaves were simple, non-petiolated and inserted in a fan shape, wrapping the stem. The blade was symmetrical with an ensiform (falcate) shape, succulent texture, glabrous surface and green color. The average leaf length was 46.5/5.6 cm, and the width was 4.9/0.7 cm (mean/SD). The apex was acuminate, the base was truncate and the margin was slightly serrate. The venation pattern was parallel (Figure 1).

Microscopical inspections were conducted on fresh and dry leaves. In fresh leaves, polygonal and elongated epidermal cells and two bundles of conductive vessels were observed (Figure 2i). Paracytic or parallel stomata were detected in an enlarged view (Figure 2ii). Tabular cells followed by an amyliferous reserve parenchyma were observed in the cross section at the mesophyll level (Figure 2iii). Inside, acute crystals and fibers were observed (Figure 2iv–vi). The microscopic structures observed in the *X. caeruleum* leaves, such as epidermal cells, paracytic stomata and crystals, were described for the first time in the present study and make a new contribution to the micro-morphological characterization of this plant species.

### 2.2. Physicochemical Parameters of the Leaves and Extracts of X. caeruleum

Some quality parameters of the plant material are listed in Table 1. Under our experimental conditions, total water-soluble and HCl-insoluble ashes showed significant differences among the extracts. Ashes values were higher for the aqueous extract from the leaves of plants collected in the flowering stage (FE) than the one obtained from plants collected in the vegetative stage (VE). Insoluble ashes values for FE and VE were higher than the limits established in the literature. These results could be associated with the availability of minerals in the soil, but the quantification of heavy metals needs to be addressed in future studies. The moisture content was similar, about 11% for VE and 10.5% for FE, while no differences were found for water- or ethanol-soluble constituents. The highest extraction rates were obtained with water or 30% ethanol.

Taking into consideration the traditional use of *X. caeruleum* and the yield of water-soluble constituents, the aqueous extract was selected for further studies and its physicochemical parameters were evaluated. Total solids, refractive index and relative density did not show significant differences between the extracts (Table 2). However, VE was slightly more acidic than FE. The acidity of the extracts could be associated with the presence of phenolic compounds. As expected, the similarity observed for total solids between the extracts was in agreement with the soluble constituents. Refractive index and relative density showed no differences between VE and FE. These parameters are characteristic for each drug and its extraction procedure; therefore, the phenological stage did not lead to significant differences.

The physicochemical parameters of the *X. caeruleum* leaves and its aqueous extract were significantly different for the two phenological stages, which can affect the pharmacological activity of the extracts. These results were reported here for the first time for this species.

### 2.3. Phytochemical Study

#### 2.3.1. Phytochemical Screening

Color and precipitation reactions using appropriate reagents suggested the presence of alkaloids, reducing sugars, phenolic compounds, coumarins, terpenoids and flavonoids. The foaming assays were negative, thus indicating the absence of saponins in the extracts.

#### 2.3.2. Thin Layer Chromatographic (TLC) Profile

The TLC profile showed one fluorescent spot under UV light at 254 nm at the application point (see Appendix A), indicative of chromophore groups. Several other yellow, light blue and reddish-brown spots were observed along the chromatographic plate under UV light at 366 nm (see Appendix A). Purple, light pink and reddish-brown spots were observed after revealing the plate with anisaldehyde (see Appendix A). The TLC profile suggested the presence of flavonoids, coumarins, terpenoids and other phenolic compounds.

#### 2.3.3. Ultraviolet (UV) and Infrared (IR) Profiles

Both extracts showed similar UV and IR profiles. In the UV profile, two bands were observed at 235 and 249 nm (see Appendix A), indicative of the presence of phenolic compounds. In the IR profile, bands were observed in the range of 3500–3200 cm^−1^, related to the presence of phenolic-type OH groups, and in the range of 1600–1500 cm^−1^, related to aromatic C=C bonds (see Appendix A), which corroborated the presence of phenolic compounds within the extracts.

#### 2.3.4. HPLC/DAD Profile

High-performance liquid chromatography analysis was carried out on the aqueous extracts, and several standard reference substances were used to identify some flavonoids and phenolic acids. Figure 3 displays the chromatogram of the VE extract.

The chromatogram shows an abundance of major peaks between 4.4 and 5.9 min, which was associated with the presence of polar compounds, mainly phenolic acids, according to the UV spectrum of each peak. This was an expected result, taking into account that the analysis was carried out on the aqueous extract of the leaves.

By comparing the retention times in the chromatogram with the ones of the standards (Figure 4), seven compounds could be identified. Gallic acid was identified as one of the major peaks, while some flavonoids and phenolic compounds were also present. These were: catechin, caffeic acid, ferulic acid, kaempferol 3-*O*-glucoside, cinnamic acid and quercetin. The presence of a single peak in each chromatogram was indicative of the standards’ purity and provided validation for peak identification.

#### 2.3.5. Total Phenols and Flavonoids Content

The VE aqueous extract from the *X. caeruleum* leaves (vegetative stage) showed the highest phenol and flavonoid content (Table 3), which could be related to the differences found in the pH values when measuring the physicochemical parameters of the extracts.

### 2.4. Diuretic Activity

The diuretic activity of the VE and FE aqueous extracts from the *X. caeruleum* leaves was evaluated and compared to that of furosemide, hydrochlorothiazide and spironolactone, which were well established diuretic drugs. The rats from the G.5 (vegetative) and G.6 (flowering) groups showed a significant increase (*p* < 0.05) in the urinary flow and the urinary levels of Na^+^ and K^+^ when compared to the negative control group (G.1) (Table 4). The VE group (G.5) showed higher values for these parameters compared to group G.6 (FE). The parameters observed for G.6 (FE) resembled those of the spironolactone (G.3) group, but the VE group (G.5) showed similar values for urinary flow and Na^+^ concentration as group G.4, which received furosemide. Interestingly, the urinary K^+^ concentration in group G.5 (VE) was similar to that in the hydrochlorothiazide group (G.2). The Lipschitz Index for the VE group (G.5) was similar to the furosemide group (G.4); by contrast, the FE group (G.6) was similar to the spironolactone group (G.3) (Table 4).

In summary, diuretic activity was demonstrated for both extracts, thus providing evidence to corroborate the traditional use of this plant species in Cuba. Interestingly, the diuretic activity was higher for the aqueous extract of the leaves of *X. caeruleum* collected in the vegetative stage (VE) than in the flowering stage (FE). This was the first pharmacological study made on this species.

### 2.5. Acute Oral Toxicity

Animals showed no signs of mortality or toxicity during the 14 days following a single administration of the extract at a dose of 2000 mg/kg b.w. Body weight increased over time, which is a parameter indicative of low or no toxicity (data not shown). No lesions were found upon macroscopic examination of the liver, kidneys, heart, lungs and spleen in any animal. Therefore, the extracts of *X. caeruleum* leaves were classified as category 4 according to the Globally Harmonized System (GHS) [24]. These results suggested that the extracts of *X. caeruleum* leaf have no acute toxicity after oral administration under our experimental conditions.

## 3. Discussion

The present study compared, for the first time, the quality parameters, chemical characteristics, diuretic activity and acute oral toxicity of *X. caeruleum* leaves collected in the vegetative and flowering stages and their aqueous extracts, making a novel contribution to the scientific knowledge on this species. Therefore, it provided the first evidence of the efficacy and safety of this plant species as a diuretic.

It is known that the pharmacological activity of medicinal plants is closely related to their phytochemical constituents [25]. In addition, the presence of secondary metabolites in a given plant can be a function of abiotic (climate, geology and others), biotic (predators and parasites) and genetic (species, variety and cultivar) factors. The harvest conditions (hygrometry, location and others) and the phenological stage of the plant should also be points of attention [26]. The phytochemical composition of medicinal plants is very complex; therefore, qualitative and quantitative parameters are measured for the quality control of herbal drugs, extracts and phytomedicines, guaranteeing their safety and efficacy [27]. In this context, the present research reported a comparative study of the pharmacognostic parameters of the *X. caeruleum* leaves collected during two phenological stages.

The macroscopic and microscopic characteristics of the leaves were similar for the two studied phenological stages. Macroscopic characteristics matched with the results reported in the literature [28], which confirmed the identity of the plant. Quality parameters of the leaves have been proposed to confirm the purity and identity of the plant material and to avoid adulteration or substitution of raw materials [29]. Previously, microscopic analysis for this species reported the distribution of phenylphenalenones in the roots, stems and leaves by histochemical and spectroscopic methods [30,31,32]. To our knowledge, epidermal cells, paracytic stomata and crystals were reported here for the first time in the leaves of *X. caeruleum*.

The higher content of ashes (total soluble in water and insoluble in HCl) observed in FE compared to VE can be explained as follows: During flowering, plants require higher amounts of minerals than in the vegetative stage in order to guarantee reproduction and species conservation [33], and ashes are indicative of the presence of inorganic compounds [29]. The content of HCl-insoluble ashes (9.5% for VE and 11.6% for FE) was higher than the 2% limit established in the Chinese Pharmacopeia [34]. These results might be associated with the availability of nutrients in the soil. The determination of ashes is accepted for assessing the purity and identification of drugs, and the HCl-insoluble ash content is used as a measure for the presence of heavy metals and silica crystals in the drug [35]. Heavy metals can be toxic [29]; therefore, heavy metal quantification should be carried out in future studies.

The moisture content was similar, about 11%, in both plant stages, while no differences were found for soluble constituents. Moisture content below 14% is considered important for avoiding the growth of microorganisms and drug degradation [29]. The highest extraction rates of secondary metabolites were produced by a 30% ethanol-water mixture and by 100% water. Water decoction is the most widespread extraction method in the traditional use of *X. caeruleum* [8,9,10,11,12,13,14]. Therefore, the aqueous extracts (VE and FE) were compared with regard to other physicochemical parameters, including preliminary phytochemical composition, total phenol and flavonoid content, diuretic activity and acute oral toxicity.

The urinary concentration of potassium in the VE group was lower than that in the furosemide group, but it was similar to that in the hydrochlorothiazide group, which was interesting because high values of K^+^ in urine have been associated with several adverse effects [36]. By contrast, all evaluated parameters showed similar values for the FE and spironolactone groups. Potassium is one of the most important electrolytes in the human body because it is essential for the cell-membrane potential, especially in the heart, muscles and nerve tissues. Therefore, potassium depletion is related to several human diseases. The main adverse effects of furosemide are associated with the loss of potassium [37]. The VE extract showed aquaretic and natriuretic activity similar to furosemide, but the kaliuretic activity was reduced in a fashion comparable to hydrochlorothiazide, thus reducing the possibility of adverse effects. The presence of flavonoids can be associated with the diuretic activity of the plant extracts because flavonoids increase the activity of prostacyclin synthase, leading to the release of renal prostaglandin, which in turn has been associated with diuresis [38], regional blood flow stimulation, vasodilatation and the inhibition of the tubular reabsorption of water and anions [39]. Therefore, the higher content of phenols and flavonoids observed in the VE extract might explain the differences observed for the diuretic activity between FE and VE.

The urinary flow, the concentration of sodium and the Lipschitz index showed similar values in rats receiving VE and furosemide but were higher than those observed in the control group. It has been reported that isoquercetin, catechins and caffeine seem to be associated with the ethnomedicinal use as a diuretic of extracts from *Tropaeolum majus* and *Eleusine coracana Linn* [40,41]. These authors propose a mechanism involving the inhibition of Na^+^/K^+^-ATPase, which may explain the increase in K^+^ and Na^+^ in the urine of rats receiving VE and FE. Gallic acid isolated from *Mimosa bimucronata* (DC.) leaves augments urinary volume excretion and induces diuresis and saluresis (Na^+^ and Cl^-^) in rats but without interfering with K^+^ excretion [42]. This suggested a mechanism of action different from the one related to VE and FE, which in turn may be related to the combined pharmacological effects of the different constituents present in complex plant extracts [3]. Further research is necessary in order to elucidate the mechanism of action of *X. caeruleum*. Finally, the extracts did not show acute oral toxicity at a dosage of 2000 mg/kg, which corroborated ethnomedical claims that this plant species should be considered ‘not harmful’ [12,13].

By taking into account that ash content (total water-soluble and HCl-insoluble) and pH were significantly lower in FE than in VE, which was desirable for a new phytochemical formulation, and the rest of the studied variables (total solids, refractive index and relative density, oral acute toxicity) did not show significant differences between the extracts, we selected VE as the most suitable for further evaluation and extract component identification. The presence of some flavonoids (i.e., quercetin, catechin and kaempferol-3-*O*-glucoside) and phenolic acids (i.e., gallic, ferulic, caffeic and cinnamic acids) was reported in this species for the first time. Other flavonoids such as apigenin and its glucosides have been reported to be present in *X. caeruleum* [16,18]. Xiphidione has been identified as a major compound [43], and phenylphenalenone-related compounds have been proposed as chemotaxonomic markers for this species [17]. The presence of phenolic compounds in an extract has been associated with a reduction in pH [44]. Our data supported this hypothesis since the total phenol and flavonoid content was higher in VE than in FE. *X. caeruleum* plants remain in the vegetative stage for longer time periods than in FE, thus providing higher drug availability. The fact that diuretic activity was higher in VE than in FE also supported our selection that VE seems to be a more suitable candidate for developing future pharmaceutical formulations.

This is the first study to compare the effect of phenological stages on the pharmacognostic parameters, diuretic activity and acute oral toxicity of *X. caeruleum* and it is the first biological study on this species. The differences found in the diuretic activity in relation to the phenological stage of the plant were interesting and should stimulate additional research. This information must be taken into consideration during the collection of the plant material, either for traditional use or for the production of phytomedicines. Further studies should focus on the identification of other phytochemical constituents of the extract in order to understand the observed differences. In addition, these results should be divulged to the Cuban population to promote the rational use of medicinal plants and stress the importance of the phenological stage of this species for the preparation of traditional remedies.

## 4. Materials and Methods

### 4.1. Solvents and Chemicals

All solvents used were of analytical grade and were purchased from Merck (Darmstadt, Germany). The standards for quercetin (CAS 849061-97-8, purity > 95%), gallic acid (CAS 5995-86-8, purity > 99%), caffeic acid, catechin (CAS 331-39-5, purity > 98%), ferulic acid (CAS 537-98-4, purity: 99%), kaempferol 3-*O*-glucoside (CAS 480-10-4, purity > 96%) and cinnamic acid (CAS 140-10-3, purity: 99%), as well as the Folin–Ciocalteu reagent were purchased from Sigma-Aldrich (Saint Louis, MO, USA). HCl was provided by UNI-Chem (Zhejiang, China), and safranin was acquired from Panreac (Barcelona, Spain). Furosemide, hydrochlorothiazide and spironolactone were obtained from Laboratorios MedSol (Havana, Cuba).

### 4.2. Plant Material

Thirty specimens of *X. caeruleum* Aubl. were collected in March and November 2018 at Ceiba del Agua (22°52′12″ N; 82°37′48″ W), Artemisa, Cuba, respectively, plants were in vegetative or flowering phenological stages. They were identified by MSc. José Ángel García Beltrán in the National Botanical Garden of Cuba, where a voucher specimen (HFC 90278) was deposited. The leaves were separated, manually cut into small pieces with the help of scissors and dried in an oven at 40 °C for six days.

### 4.3. Morphological Characteristics

The macroscopic characteristics of twenty leaves were determined by visual inspection. The length and width of 100 leaves were measured with the help of a graduated ruler. Leaves were decolorized with 10% (*v*/*v*) sodium hypochlorite, colored with 1% (*v*/*v*) safranin solution in water and fixed with glycerin for observing the microscopic characteristics, as reported elsewhere [45]. Observations were made with a camera-coupled optical NXZ-N107 microscope (Novel, Ningbo, China).

### 4.4. Physicochemical Parameters of the Leaves

Residual humidity, total ashes, soluble ashes in water, insoluble ashes in 10% HCl, soluble constituents in water, 30% ethanol, 50% ethanol and 80% ethanol were determined. The residual humidity was determined by the azeotropic method. All assays were performed according to methods described elsewhere [34]. In short, residual humidity (H) was determined by the water content displaced by chloroform (previously dehydrated) and heat. Ashes (total, water-soluble and HCl-insoluble) were determined as the weight of the remaining material after continuous heating at 700 °C until constant weight. Data were expressed as % in relation to the starting material, except for HCl-insoluble ashes, where results were expressed as 100% (% in relation to the starting material). Starting materials were: 2 g of drug (total), total ashes dissolved in 12 mL of H_2_O (water-soluble) or in 3 mL of 10% HCl (HCl-insoluble). Soluble constituents in water, 30% ethanol, 50% ethanol and 80% ethanol were determined as follows: 5 g of drug in 100 mL of solvent were stirred, filtered and a 20 mL aliquot was dried on a water bath until constant weight. Results were reported as (R*500*100)/(5 g *(100-H)), where R was the remains’ weight (in g) and H was the previously calculated residual humidity.

### 4.5. Preparation and Physicochemical Analysis of the Aqueous Extracts

Twenty grams of plant material were extracted by decoction in 100 mL of distillated water during 15 min. After that, the extracts were filtered and kept at 20 °C for further analysis. Organoleptic properties (color, taste, flavor and transparency), total solids, refraction index, pH and relative density of the extracts were determined according to the methods described elsewhere [34]. In short, for the determination of total solids, 5 mL of the extracts were heated in a water bath until they reached constant weight. Results were reported as % of the remains’ weight in relation to the starting volume.

### 4.6. Phytochemical Screening

Phytochemical screening was carried out by means of the assays of Dragendorff, Mayer and Wagner for alkaloids; Fehling for reducing sugars; the foaming assay for saponins; the ferric chloride assay for polyphenols; and the Shinoda assay for flavonoids. All assays were performed according to the procedures described elsewhere [46]. In short:

Assays for alkaloids: 6 mL of the aqueous extract was acidified with HCl, and Mayer, Wagner or Dragendorff reagents were added. The formation of a cream (Mayer) or reddish-brown precipitate (Dragendorff and Wagner) was considered positive for the presence of alkaloids.

Assay for reducing sugars: 4 mL of a 1:1:2 mixture of Fehling A, Fehling B and aqueous extract was heated for 10 min. The formation of a red precipitate was considered positive for reducing sugars.

Saponin assay: the formation of foam after vigorously shaking the extract for 3 min was considered positive for the presence of saponins. The foam that forms should last for 2 min or more.

Assay for phenols: a few drops of a neutral 5% ferric chloride solution were added to 2 mL of aqueous extract. The development of a dark green color was considered a positive indicator of the presence of phenolic compounds.

Test for flavonoids (Shinoda assay): A few fragments of magnesium ribbon and concentrated HCl were added to the extract. The appearance of red to pink colors after a few minutes indicates the presence of flavonoids.

### 4.7. Chromatographic and Spectroscopic Profiles of the Aqueous Extracts

The TLC profile was established on silica gel F254 plates (Merck, Darmstadt, Germany) using butanol/acetic acid/water (65:25:10) as the mobile phase. UV light at 254 nm and 366 nm, visible light anisaldehyde and heat were used to detect spots. The ultraviolet spectra of the extracts were recorded from 200 nm to 400 nm on a SPECORD-200 plus UV-Vis spectrophotometer (Analytik Jena, Jena, Germany). For IR spectroscopy, a KBr tablet containing the dried extract was prepared and the IR spectrum was recorded from 4000 to 700 cm^−1^ on a FT/IR-460 spectroscope (Jasco plus, Oklahoma, OK, USA).

### 4.8. HPLC/DAD Profile

The analysis was carried out just on the VE aqueous extracts at 10 mg/mL. The samples were filtered through a 45 μm filter to remove impurities, and they were directly injected into the equipment. An Agilent Technologies liquid chromatograph 1200 (Santa Clara, CA, USA) was used for the study.

Several reference substances were used as standards, more particularly the flavonoids rutin, quercetin, kaempferol-3-*O*-glucoside, apigenin, catechin, epicatechin and phenolic acids (ferulic, caffeic, cinnamic and gallic acids, 98% purity, Sigma-Aldrich, St Louis, MO, USA).

The separations were carried out on a Kinetex EVO C18 (5 µm, 250 mm × 4.6 mm) column. The elution was carried out at a flow rate of 1 mL/min, the injection volume was 20 μL, and a pressure of 25 μPa was used. The oven temperature was 25 °C. The detection was achieved with a diode array detector (DAD) in the range 200–400 nm. The mobile phase consisted of water with 0.1% formic acid (solvent A) and acetonitrile-methanol (7:3) with 0.1% formic acid (solvent B), using a gradient program of 5% B in 0–5 min, 5–100% B in 5–50 min, 100% B in 50–60 min and 5% B in 60–65 min for re-equilibration.

### 4.9. Total Phenol and Flavonoid Content

The total amount of phenolic compounds was determined by the Folin–Ciocalteu reagent [47]. A total of 200 μL of extract was dissolved in 10 mL of Folin–Ciocalteu solution and 8 mL of saturated sodium carbonate solution. After 90 min, the absorbance was recorded at 765 nm with a SPECORD-200 Plus UV–Vis spectrophotometer (Analytik Jena, Jena, Germany). Gallic acid was used as a standard. A calibration curve Y = 0.0131X + 0.0240 (R^2^ = 0.9987) was obtained (see Appendix A).

The total amount of flavonoids was determined using the aluminum trichloride (AlCl_3_) reagent [47]. A volume of 1.5 mL (1 mg/mL) of extract was added to an equal volume of a 2% AlCl_3_ solution. The mixture was vigorously shaken, and the absorbance was recorded at 367 nm after 10 min of incubation with a SPECORD-200 Plus UV–Vis spectrophotometer (Analytik Jena, Jena, Germany). Quercetin was used as a standard. A calibration curve Y = 0.0049X + 0.0167 (R^2^ = 0.9954) was obtained (see Appendix A).

### 4.10. Animals

Wistar rats (200–250 g body weight) were purchased from CENPALAB (Havana, Cuba) and kept in collective cages at 22 °C under a 12 h light/dark cycle (lights on at 07:00 h) with ad libitum access to laboratory food and tap water. The investigations followed Cuban laws for preclinical studies and were in agreement with international regulations for animal care. Protocols were approved by the Ethical Committee of the Institute of Basic and Preclinical Sciences “Victoria de Girón”, Medical University of Havana, Cuba (approval code 28-2019, December 2019).

### 4.11. Diuretic Activity

Diuretic activity was evaluated in Wistar rats using metabolic cages. Thirty-six male animals were randomly divided into six groups of equal size (*n* = 6), as follows:Group 1: Physiological solution of sodium chloride, 0.9% (negative control)Group 2: Hydrochlorothiazide in CMC (10 mg/kg) (positive control)Group 3: Spironolactone in CMC (10 mg/kg) (positive control)Group 4: Furosemide in CMC (20 mg/kg) (positive control)Group 5: Aqueous extract of *X. caeruleum* (vegetative) in CMC (400 mg/kg)Group 6: Aqueous extract of *X. caeruleum* (flowering) in CMC (400 mg/kg)

All treatments were prepared as a suspension in carboxymethyl cellulose (CMC) and administered by gastric gavage using a volume of 3 mL per rat. Afterward, rats were placed in the metabolic cages, and the excreted urine volume was measured every hour for four hours. The urinary flow and Lipschitz index [48] were calculated, and urinary concentrations of sodium and potassium were determined by flame photometry (Corning 400, San Francisco, CA, USA).

### 4.12. Acute Oral Toxicity

In vivo acute oral toxicity of the aqueous extract from the *X. caeruleum* leaves was investigated according to OECD Guideline 423, consisting of a single-dose plus 14-day observation study. The Globally Harmonized System (GHS) was used for classification [24]. Three male or three female Wistar rats were randomly distributed per group. One group of male and one of female animals received a single dose of 2000 mg/kg per body weight of each extract, while the control group received just water. After treatment, the following toxicity signs were recorded: changes in skin and fur, eyes, respiration, behavioral pattern and mortality in all animals. Observations were made at 0.5, 1, 6 and 24 h after treatment; afterward, signs were recorded daily for 14 days, while body weight was recorded weekly. At day 15, all animals were anesthetized by petroleum ether inhalation, sacrificed by cervical dislocation and their internal organs were observed [49].

### 4.13. Statistical Analysis

The mean and standard deviation were determined for all variables. The student t-test was used to compare the content of phenols and flavonoids, as well as the physicochemical parameters of the leaves and aqueous extract. Biological variables were compared by means of One-Way Analysis of Variance (ANOVA) with Student–Newman-–Keuls post hoc test. A *p* value less than 0.05 was considered to be statistically significant.

## 5. Conclusions

Some microscopic structures, such as epidermal cells, paracytic stomata and crystals, were described for the first time in *X. caeruleum* leaves. Aqueous extracts from *X. caeruleum* leaves showed no toxicity signs in rats after receiving a single dose of 2000 mg/kg per body weight under our experimental conditions. Flavonoids and phenols present in the VE and FE extracts might explain at least in part the traditional use as a diuretic. Plants harvested during the vegetative phenological stage contained higher levels of phenols and flavonoids and showed more pronounced diuretic activity than the ones collected in the flowering state. However, differences in polyphenol profiles between VE and FE extracts suggested that more detailed phytochemical, pharmacological and toxicological investigations should be carried out to standardize the harvesting and extraction conditions in order to use *X. caeruleum* leaf extracts as herbal medicines.

## Figures and Tables

**Figure 1 plants-12-01268-f001:**
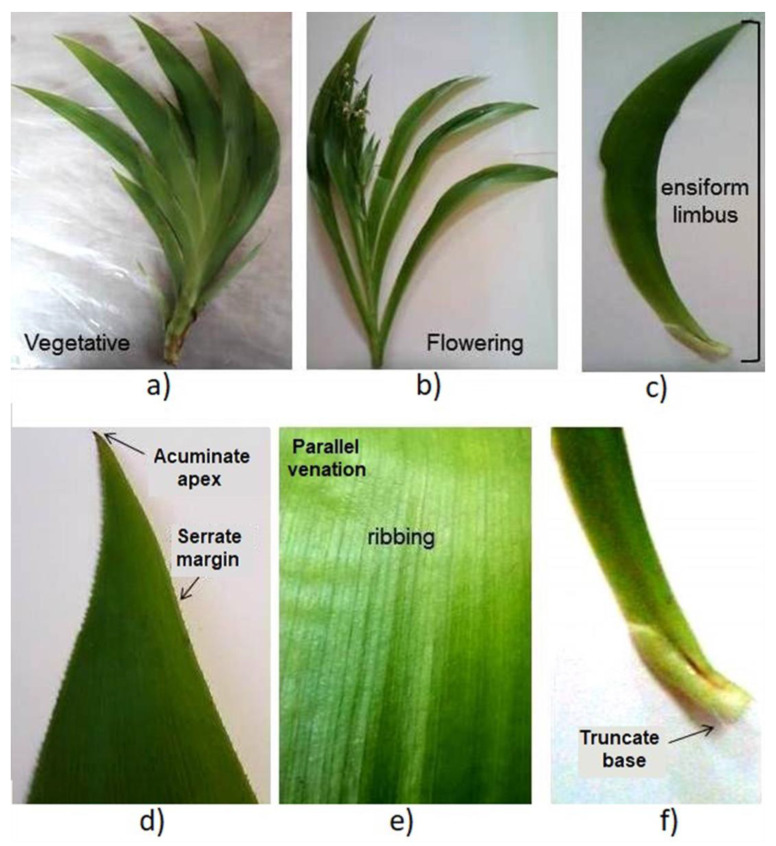
Macroscopic characteristics of the *X. caeruleum* leaves. (**a**): vegetative stage, (**b**): flowering stage, (**c**): blade, (**d**–**f**): magnified portions of the leaves, apex and margin (**d**), venation pattern (**e**) and base (**f**).

**Figure 2 plants-12-01268-f002:**
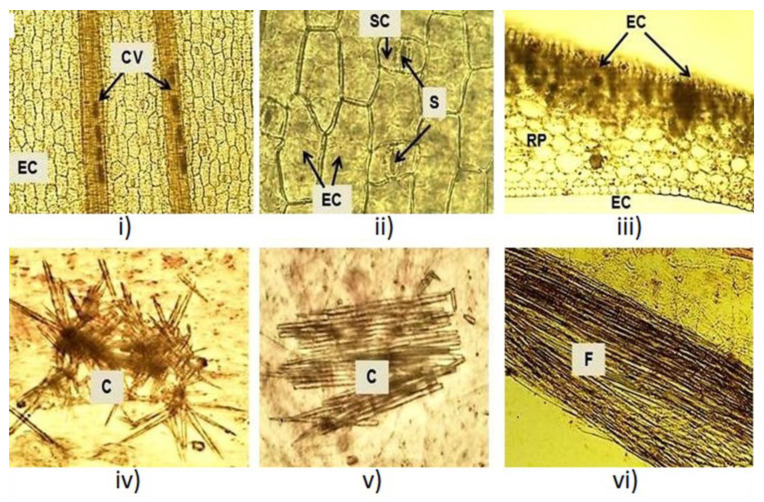
Microscopic characteristics of the *X. caeruleum* leaves. CV: Conductive vessels; EC: Epidermal cells; S: Stomata; RP: Reserve parenchyma; C: Crystals; F: Fibers.

**Figure 3 plants-12-01268-f003:**
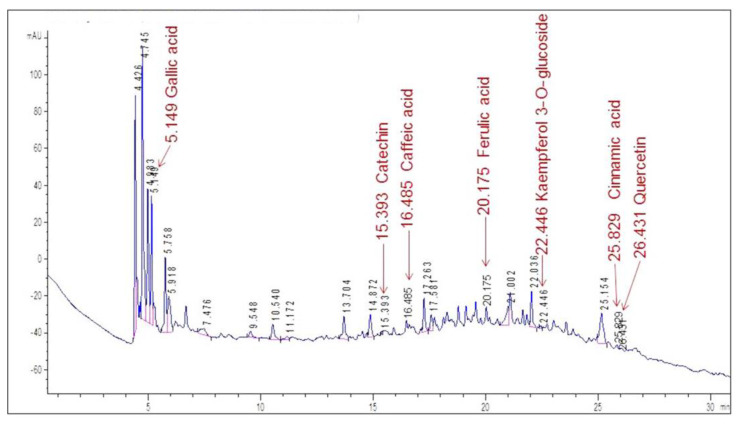
HPLC/DAD profile of the VE aqueous extract of *X. caeruleum* leaves. Labels indicate the retention time for each identified compound.

**Figure 4 plants-12-01268-f004:**
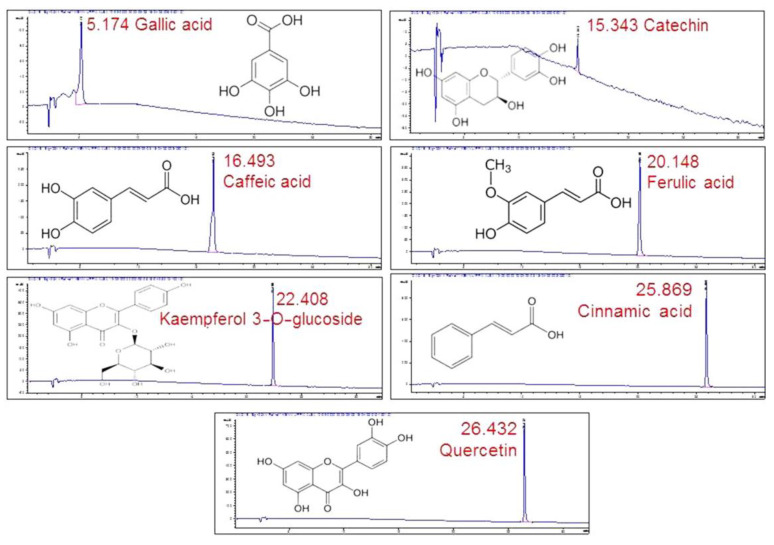
HPLC/DAD profiles of the standards used in the study with their retention times.

**Table 1 plants-12-01268-t001:** Physicochemical parameters of the *X. caeruleum* leaves collected in vegetative (VE) and flowering (FE) stages.

Parameter	VE (%) ^1^	FE (%) ^1^
Total ashes *	13.12/0.04	14.25/0.39
Water-soluble ashes *	8.26/0.04	10.85/0.51
Acid-insoluble ashes (in HCl) *	9.50/0.11	11.60/0.32
Moisture content	11.00/1.41	10.50/0.70
Water-soluble constituents	19.46/0.17	19.77/0.53
Ethanol 30% soluble constituents	19.25/0.61	18.99/1.34
Ethanol 50% soluble constituents	12.89/0.24	12.66/0.15
Ethanol 80% soluble constituents	9.93/0.37	10.10/0.35

^1^ Values are expressed as mean/SD. *: significant differences between the two phenological stages (t-student test, *p* < 0.05).

**Table 2 plants-12-01268-t002:** Physicochemical parameters of the aqueous extract from the *X. caeruleum* leaves collected in vegetative (VE) and flowering (FE) stages.

Parameter	VE ^1^	FE ^1^
pH *	5.95/0.02	6.20/0.01
Total solids (%)	2.74/0.06	2.91/0.24
Refractive index	1.3314/0.0003	1.3316/0.0001
Relative density (g/mL)	1.0157/0.0005	1.0167/0.0005

^1^ Values are expressed as mean/SD. *: significant differences between the two phenological stages (t-student test, *p* < 0.05).

**Table 3 plants-12-01268-t003:** Contents of flavonoids and phenolic compounds in the aqueous extract from the *X. caeruleum* leaves collected in the vegetative (VE) and flowering (FE) stages.

Parameter	VE (%) ^1^	FE (%) ^1^
Content of phenols (mg/mL) *	1.30/0.01	1.26/0.01
Content of flavonoids (mg/mL) *	0.45/0.02	0.34/0.01

^1^ Values are expressed as mean/SD. *: significant differences between the two phenological stages (t−student test, *p* < 0.05).

**Table 4 plants-12-01268-t004:** Diuretic activity of the aqueous extract of the leaves of *X. caeruleum* leaves collected in the vegetative (VE) and flowering (FE) stages ^1^.

Group	Urinary Flow	C(Na^+^)	C(K^+^)	Lipschitz
	(µLmin^−1^)	(mEq)	(mEq)	Index
G.1 NaCl 0.9% (*v*/*v*) control	3.1/0.6 ^a^	46.0/7.3 ^e^	15.7/6.1 ^i^	-
G.2 Hydrochlorothiazide	4.8/0.5 ^b^	74.6/3.1 ^f^	33.3/5.2 ^j^	1.34
G.3 Spironolactone	6.0/0.6 ^c^	62.0/1.7 ^g^	20.7/5.0 ^k^	1.78
G.4 Furosemide	7.0/0.5 ^d^	84.5/298 ^h^	55.0/5.8 ^l^	2.06
G.5 VE extract	6.8/0.4 ^d^	85.7/2.5 ^h^	33.0/6.3 ^j^	1.96
G.6 FE extract	5.8/0.6 ^c^	63.7/4.7 ^g^	23.4/5.2 ^k^	1.72

^1^ The values are expressed as mean/SD. Different letters indicate significant differences among groups (*p* < 0.05, ANOVA and post-hoc Student–Newman–Keuls test).

## Data Availability

Not applicable.

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
