# Peer review of "Pharmacognostic Study, Diuretic Activity and Acute Oral Toxicity of the Leaves of Xiphidium caeruleum Aubl. Collected in Two Different Phenological Stages"

_plants, 2023, doi:10.3390/plants12061268_

Round 1
Reviewer 1 Report (Previous Reviewer 1)
Most of the comments have been addressed, and the manuscript has been carefully revised.
1, As the results of TLC, UV and IR only provide limited information, it is suggested to provide the related figures (Figure 3 and Figure 4) in the supplementary materials.
2, The quality (resolution) of figures need to be improved.
3, The HPLC chromatogram of the mixture of reference chemical compounds needs to be provided in Figure 5.
4, The contents of these identified compounds with reference chemical compounds can be determined in the plants’ extracts.
5, The purity of the chemical compounds should be indicated in “4.1. Solvents and chemicals”.
Author Response
Response to reviewer 1
comments
Point 1. English language and style are fine/minor spell check required
Response 1. Multiple typos and stile errors were corrected across the manuscript by a native English speaker.
Point 2. Results can be improved in order to be clearly presented
Response 2. Results and discussion sections were carefully reviewed. Some duplicity in conclussion and references were removed. The HPLC/DAD profile of the standards used in the study was included which allowed us to report the Retention times, width and height of the identified peaks at the HPLC profile of the whole extract, as well as the relative abundance of the identified compounds within the extract.
Point 3. As the results of TLC, UV and IR only provide limited information, it is suggested to provide the related figures (Figure 3 and Figure 4) in the supplementary materials.
Response 3. Figures 3 and 4 were removed from the manuscript and they were inserted in the support information file as Figures S.1 and S.2.
Point 4. The quality (resolution) of figures need to be improved.
Response 4. The quality of each figure was enhanced, thus providing 600 dpi resolution for a 8.5 inch figure width.
Point 5. The HPLC chromatogram of the mixture of reference chemical compounds needs to be provided in Figure 5.
Response 5. They were provided as the new figure 3.
Point 6. The contents of these identified compounds with reference chemical compounds can be determined in the plants’ extracts.
Response 6. They were calculated and said information is reported in the new Table 3.
Point 7. The purity of the chemical compounds should be indicated in “4.1. Solvents and chemicals”.
Response 7. The purity of the chemical compounds is now reported in section “4.1”. Lisset Herrera Isidrón Corresponding author
Reviewer 2 Report (Previous Reviewer 3)
The authors have made the indicated modifications and the article has improved substantially. For this reason, I consider that the article can be considered for publication in Plants journal in its current form.
Author Response
Response to reviewer 2 comments
Point 1. English language and style are fine/minor spell check required
Response 1. Multiple typos and stile errors were corrected across the manuscript by a native English speaker.
Point 2. Methods can be improved in order to be adequately described
Response 2. More information, specifically the purity of the chemical compounds used in this study was indicated in section “4.1. Solvents and chemicals”.
Lisset Herrera Isidrón
Corresponding author
Round 2
Reviewer 1 Report (Previous Reviewer 1)
The manuscript may be accepted for publication in present form.
Author Response
Response to reviewer 1 comments
Point 1. English language and style are fine/minor spell check required
Response 1. Multiple typos and style errors were corrected across the manuscript by a native English speaker.
Lisset Herrera Isidrón
Corresponding author

This manuscript is a resubmission of an earlier submission. The following is a list of the peer review reports and author responses from that submission.
Round 1
Reviewer 1 Report
In this study, the pharmacognostic parameters and preliminary phytochemical composition of Xiphidium caeruleum are determined, as well as the diuretic activity and acute oral toxicity of the aqueous extracts from this plant collected in vegetative and flowering stages are evaluated. Although this study can provide some information for the study of X. caeruleum, it is a preliminary study. Followings are some suggestions for further revisions.
1, Some further study can be supplemented, for example, the phytochemical analysis using hyphenated techniques such as LC-MS; further fractionate the water extract and test their pharmacological activity to prove which group of compounds may exert the actions.
2, In the abstract, there are not experimental results to support the conclusion that “the diuretic activity was attributed to the presence of flavonoids and phenols in the extracts”.
3, In the introduction section, more detail information about Xiphidium caeruleum can be introduced.
4, The section “2.3. Physico-Chemical Parameters of the Aqueous Extracts” can be merged into section “2.2. Physico-Chemical Parameters of X. caeruleum Leaves”.
5, The quality of figures (such as Figure 4) can be improved.
Author Response
Authors are very grateful for your comments; changes were made accordingly. It was not possible to include the LC/MS and GC/MS chromatographic profiles. However, we consider that the novel pharmacognostic, pharmacological and toxicological data provided here should be of interest for the readers of Plants. Detailed explanation is disclosed below.
Kind regards,
Lisset Herrera Isidrón
Corresponding author
Response to reviewer 1 comments
- Some further study can be supplemented, for example, the phytochemical analysis using hyphenated techniques such as LC- MS; further fractionate the water extract and test their pharmacological activity to prove which group of compounds may exert the actions.
Answer. We agreed with reviewer’s comment, these analyses should provide more information on which group of compounds may exert the actions and it also might reveal some insight on the extract action mechanisms. We also agreed that current data do not provide evidence enough to support that the observed diuretic activity be attributed to presence of polyphenols. In accordance, proper modifications have been introduced in the abstract and discussion sections. More detailed fractioning and phytochemical characterization will be addressed in further studies.
- In the abstract, there are not experimental results to support the conclusion that “the diuretic activity was attributed to the presence of flavonoids and phenols in the extracts”.
Answer. Same as 1. Conclusion was changed to “Flavonoids and phenols presence in VS and FS extracts provided some rationale to reported ethno-medical use as diuretic” which in our opinion it is supported by results.
- In the introduction section, more detail information about Xiphidium caeruleum can be introduced.
Answer. We agreed with reviewer’s comment, therefore, more detailed information have been included in the introduction section.
- The section “2.3. Physico-Chemical Parameters of the Aqueous Extracts” can be merged into section “2.2. Physico-Chemical Parameters of X. caeruleum Leaves”.
Answer. Former sections 2.2 and 2.3 were merged into new section 2.2.
- The quality of figures (such as Figure 4) can be improved
Answer. High quality figures (specifically new Figure 4) were included in the corrected version of the manuscript.
In addition, following highlights support paper novelty and justify that useful data is provided for rising interest to the readers of Plants:
- Differences in total phenols and flavonoids content was reported between extract from plant in flowering and vegetative phaenological stages. Said differences matched with the observed differences in diuretic activity.
- In our knowledge, there is not previous scientific information on caeruleum pharmacological properties.
- Epidermal cells, stomata and crystals on the leave surface were reported here for the first time.
Reviewer 2 Report
Dear Authors,
The present study evaluates the diuretic activity and acute oral toxicity of the aqueous extracts of X. caeruleum. The research subject is interesting and brings scientific important data in the field, as it deals with a subject that is currently of great interest. Some changes of the manuscript should nevertheless be performed in order to improve its quality. Following specific changes should thus be performed:
Minor changes
Line 32: vegetative extract?
Please add paternity of the species, the first time it appears in the text.
Major changes
Abstract: Please be specific about “well-established methods” and about the species (which part of it?) collected “in vegetative and flowering stages”. I particularly do not understand the difference between these 2 stages.
Introduction: This section should contain information regarding similar existing studies in literature and, in comparison, authors should emphasize the novelty and originality of their study. Please explain and cite each of these studies. Background of the study is also not clear. The purpose of the study needs to be rephrased to become clearer. Also, you have little information about the species that represents the subject of your study (why is it known, for which purposes/medicinal uses, why did you choose it etc.). Please add further information and justifications and modify accordingly. It is absolutely clear that the study is not well documented as too little references are cited. This section needs serious changes, it is too short and does not offer a proper introduction into the large context. Please offer a rationale for choosing this species and offer more details about it in the context of your study.
Results and Discussions: Here you should emphasize novelty and originality of the present study once again. You compare your results with the ones obtained by other authors, but you need to highlight what you bring in novelty compared to these. I am sure that all the assays you performed on the species are valuable and bring important and interesting results, but in the context of your study and taken into consideration you deal with a lesser studied species, it would have been better to add the phytochemical analysis of extracts, rather than perform all these assays. This could help make correlations between the tested biological activity and these compounds. in the Discussions, you mention some compounds but it is not clear if they were evaluated by you or other authors. Clearly you need to offer more details and develop this part in order to bring consistency to your study. In the present form, it is not correctly justified.
Conclusions: Please offer perspectives of your study.
All these suggested changes should be performed in order to bring further improvements to the manuscript.
Author Response
Authors are very grateful for your comments; changes were made accordingly. Detailed explanation is disclosed below.
Kind regards,
Lisset Herrera Isidrón
Corresponding author
Response to reviewer 2 comments
Minor changes
- Line 32: vegetative extract? Please add paternity of the species, the first time it appears in the text.
Answer. Vegetative extract is the one obtained from the leaves of Xiphidium caeruleum collected at vegetative stage. However, we agreed that the phrase “vegetative extract” is misleading. Therefore, we replaced it by “plant collected in vegetative stage (VS)”. The paternity of X. caeruleum was included in the introduction section.
Major changes
- Abstract: Please be specific about “well-established methods” and about the species (which part of it?) collected “in vegetative and flowering stages”. I particularly do not understand the difference between these 2 stages.
Answer. The phrase “well-established methods” was removed from the abstract and a short list of the methods used was provided. Further details were included in the Material and Methods section. The extracts were obtained from leaves, proper correction were made in the new abstract.
The life cycles of plants are divided into different stages of development (germination, vegetative, flowering, etc.), called phenological stages. As metabolites composition differed among these phenolgical stages we studied phytochemical composition of extracts obtained from plants in two different phenolgical stages: Vegetative and Flowering stages. In accordance, some literature report was included in the introduction.
- Introduction: This section should contain information regarding similar existing studies in literature and, in comparison, authors should emphasize the novelty and originality of their study. Please explain and cite each of these studies. Background of the study is also not clear. The purpose of the study needs to be rephrased to become clearer. Also, you have little information about the species that represents the subject of your study (why is it known, for which purposes/medicinal uses, why did you choose it etc.). Please add further information and justifications and modify accordingly. It is absolutely clear that the study is not well documented as too little references are cited. This section needs serious changes, it is too short and does not offer a proper introduction into the large context. Please offer a rationale for choosing this species and offer more details about it in the context of your study.
Answer. We agreed with reviewer’s comments. Introduction section was heavily modified in order to provide more information regarding previous ethnobotanical use of the plant, reported chemical composition as well as some rationale about why this species was selected and also why its diuretic activity was studied. It is important to emphasize that the scientific information about X. caeruleum is scarce, which in fact, become one of the strong point of this manuscript, several of our results were reported here for the first time.
- Results and Discussions: Here you should emphasize novelty and originality of the present study once again. You compare your results with the ones obtained by other authors, but you need to highlight what you bring in novelty compared to these. I am sure that all the assays you performed on the species are valuable and bring important and interesting results, but in the context of your study and taken into consideration you deal with a lesser studied species, it would have been better to add the phytochemical analysis of extracts, rather than perform all these assays. This could help make correlations between the tested biological activity and these compounds. in the Discussions, you mention some compounds but it is not clear if they were evaluated by you or other authors. Clearly you need to offer more details and develop this part in order to bring consistency to your study. In the present form, it is not correctly justified.
Answer. Extensive modifications were introduced in the Result and Discussion sections of the manuscript in order to compare results with reports from the literature and to highlight the paper novelty. As this is a preliminary study, we did not intend to correlate extract activity to individual components but, in accordance with reviewer’s comments we adjusted discussion and conclusions to our data.
- Conclusions: Please, offer future perspectives about your study.
Answer. Future perspectives, which included further in deep phytochemical and pharmacological studies on the extract composition, were added in the conclusions.
Reviewer 3 Report
This paper describes an initial pharmacognostic study, diuretic activity and acute oral toxicity of the leaves of Xipidium caeruleum Aubl. Collected in two different phenological stages.. The article is of interest to the scientific community, and the methods are appropriate, although I miss a study of the individual compounds present in the extracts. On the other hand, the methods as well as certain parts of the introduction and discussion need to be explained and discussed in greater depth. In any case, the work is interesting and deepens in the acknowledgement of the activity of Xipidium caeruleum.
I consider that the article is appropriate to be published in Plants journal once the authors have made major modifications to it.
Introduction: I consider the introduction very short. Delve into parts such as the chemical composition of the studied plant, previous known activities of the plant, etc.
Discussion: Address the discussion in greater depth.
Section 4: Include a "Reagent Section". In this section, authors should describe all reagents and solvents used.
Section 4.1.: Include geographic coordinates. From how many plants were leaves taken? How many leaves per plant?
Section 4.3.: The methods must be described or conveniently referenced.
Section 4.4.: How much plant is extracted? Were the extracts filtered? What temperature is "cold"?
Section 4.6.1: Describe briefly each method.
Section 4.7.1.: Include calibration curve, r2 and range of linearity of gallic acid.
Section 4.7.2.: Include calibration curve, r2 and range of linearity of quercetin.
Other minor aspects:
Line 53: Do not capitalize “phenyl….”.
Table 2,……: “mL” instead of “ml”. Unify and apply to the entire document.
Lines 272, 277, 280, 317….: Include the city and country of all the companies cited, and cite the companies of all the reagents and equipment’s employed. In case of USA companies, include the city and the state abbreviation. Unify and apply to the entire document.
Line 280: Put “-1” in superscript.
Lines 290, 291: In “AlCl3” put “3” in subscript.
Line 305: Put “n” in italics.
Line 305: Put a separation after and before “=”.
References: Put references in the correct format of the journal.
Author Response
Authors are very grateful for your comments; changes were made accordingly. Detailed explanation is disclosed below.
Kind regards,
Lisset Herrera Isidrón
Corresponding author
Response to reviewer 3 comments
I miss a study of the individual compounds present in the extracts. On the other hand, the methods as well as certain parts of the introduction and discussion need to be explained and discussed in greater depth.
Answer: As this is a preliminary study, we did not intend to carry out a full extract fractioning and individual compound identification. Introduction, methods and discussion sections received extended modifications in order to enhance clarification.
- Introduction: I consider the introduction very short. Delve into parts such as the chemical composition of the studied plant, previous known activities of the plant, etc.
Answer. We agreed with reviewer’s comment, therefore, more detailed information have been included in the introduction section.
- Discussion: Address the discussion in greater depth.
Answer. We agreed with reviewer’s comment, therefore, in depth analysis of our data has been included in the discussion section and results were contrasted with the scarce reported literature.
- Section 4: Include a "Reagent Section". In this section, authors should describe all reagents and solvents used.
Answer. Solvents and chemicals were included in new subsection 4.1. Solvents and chemicals
- Section 4.1.: Include geographic coordinates. From how many plants were leaves taken? How many leaves per plant?
Answer. Geographic coordinates and data on the number of leaves and plants used in the studio were included in new section 4.2. Plant material
- Section 4.3.: The methods must be described or conveniently referenced.
Answer. More detailed information have been included in former subsection 4.3, new section 4.4. Physico-chemical parameters of the leaves.
- Section 4.4.: How much plant is extracted? Were the extracts filtered? What temperature is "cold"?
Answer. Twenty grams of drug were extracted by decoction in one-hundred milliliters of distillated water during 15 minutes. After that, the extracts were filtered and kept at 20 0C for further analysis. Proper modifications were included in former subsection 4.4, new section 4.5. Obtention and physico-chemical parameters of the aqueous extracts.
- Section 4.6.1: Describe briefly each method.
Answer. Proper modifications were included in former subsection 4.6.1, new section 4.6. Phytochemical screening.
- Section 4.7.1.: Include calibration curve, r2 and range of linearity of gallic acid.
- Section 4.7.2.: Include calibration curve, r2 and range of linearity of quercetin.
Answer. Calibration curves with R2 coefficient and linearity range for Gallic acid and quercetin was provided in the support information, Figures S1 and S2.
- Other minor aspects:
- Line 53: Do not capitalize “phenyl….”.
- Table 2,……: “mL” instead of “ml”. Unify and apply to the entire document.
- Lines 272, 277, 280, 317….: Include the city and country of all the companies cited, and cite the companies of all the reagents and equipment’s employed. In case of USA companies, include the city and the state abbreviation. Unify and apply to the entire document.
- Line 280: Put “-1” in superscript.
- Lines 290, 291: In “AlCl3” put “3” in subscript.
- Line 305: Put “n” in italics.
- Line 305: Put a separation after and before “=”.
- References: Put references in the correct format of the journal.
Answer. All minor changes were amended in the corrected version of the manuscript
Reviewer 4 Report
The paper submitted by Gaiten et al. is focused on the evaluation of the bioactivity of extracts prepared from Xiphidium caeruleum. The biological part of the paper is well planned and all experiments seem to be correctly performed.
The major problem of the submitted paper is phytochemical part of the submitted research.
The manuscript lacks fine phytochemical screening of the extracts used for bioassays. TLC analysis is very poor and does not provide sufficient info on the content of phytochemical in the extract. The chemical screening described in section 2.4.1. is highly unspecific and does not provide substantial info on compounds present in extracts. The authors should provide HPLC screening showing what kind of compounds are presents in investigated samples. Without good phytochemical analysis the paper should not be published in Plants.
Section 4.1 should contain more info on the plant material used for the whole research: name of the expert responsible for the identification must be provided; GPS coordinates of the collection site should be given, citation for proper literature used for the authentication should be given
Author Response
Authors are very grateful for your comments; changes were made accordingly. Detailed explanation is disclosed below. HPLC screening was not provided; however, we consider that the novel pharmacognostic, pharmacological and toxicological data provided here should be of interest for the readers of Plants. Detailed explanation is disclosed below.
Kind regards,
Lisset Herrera Isidrón
Corresponding author
The authors agree with you respect to “more phytochemical information led to identify compounds in the extract, such as, LC/MS or GC/MS will improve the quality of the present research;
RESPONSE TO REVIEWER 4
The major problem of the submitted paper is phytochemical part of the submitted.
The manuscript lacks fine phytochemical screening of the extracts used for bioassays. TLC analysis is very poor and does not provide sufficient info on the content of phytochemical in the extract. The chemical screening described in section 2.4.1. is highly unspecific and does not provide substantial info on compounds present in extracts. The authors should provide HPLC screening showing what kind of compounds are presents in investigated samples. Without good phytochemical analysis the paper should not be published in Plants.
Answer. This is a preliminary study; we did not intend to carry out a phytochemical screening or to correlate extract activity to individual components. TLC, UV and IR profiles just demonstrated the presence of phenolic compounds in the extract. Further studies should be focused on the identification of more phytochemical constituents of the extract and the structure/activity relation and activity guided fractioning by using high resolution HPLC, LC/MS, GC/MS or other suitable methods.
However, this research became the first comparative study of pharmacognostic parameters, diuretic activity and acute oral toxicity of X. caeruleum at two different phenological stages. Information provided here should be taken into consideration during plant harvesting either for traditional or phytomedical uses.
In addition, following highlights support paper novelty and justify that useful data is provided for rising interest to the readers of Plants:
- Differences in total phenols and flavonoids content was reported between extract from plant in flowering and vegetative phaenological stages. Said differences matched with the observed differences in diuretic activity.
- In our knowledge, there is not previous scientific information on caeruleum pharmacological properties.
- Epidermal cells, stomata and crystals on the leave surface were reported here for the first time.
- Section 4.1 should contain more info on the plant material used for the whole research: name of the expert responsible for the identification must be provided; GPS coordinates of the collection site should be given, citation for proper literature used for the authentication should be given.
Answer. These informations were included in new subsection 4.2. Plant material.
Round 2
Reviewer 1 Report
Although the manuscript has been slightly improved, it still lack of obvious innovation. The phytochemical studies are preliminary, the results of TLC, UV and IR only provide limited information.
Reviewer 2 Report
Dear Authors,
The present study evaluates the diuretic activity and acute oral toxicity of the aqueous extracts of X. caeruleum. The authors performed most of the suggested changes after the first round of review. Following specific changes should still be performed:
Minor changes
Abstract should not “announce” starting of a new section like Background, Results...
Major changes
Introduction: You still need to make proper connections between the different concepts of this part. Please offer a rationale for choosing this species.
Results: TLC analysis does not offer an adequate separation of cpmpound, which are not even well visible as spots.
Discussions: Line 264 – you should explain how corroboration between TLC, UV and IR profiles was performed to reveal presence of flavonoids and phenols. In the present form, it is not clear and therefore, the lack of a phytochemical study is absolutely obvious. Please adequately underline how the performed analysis may sustain this lack. Please make more correlations between the tested biological activity and these compounds. Also, you should cite studies on the species performing phytochemical analysis of compounds.
Conclusions: Please state what your findings bring as innovative knowledge in scientific literature.
With all the performed changes, the manuscript reveals now some language, style and editing problems, which should be corrected.
All these suggested changes should be performed in order to bring further improvements to the manuscript.
Reviewer 3 Report
The authors have made the indicated modifications and the article has improved substantially. Anyway I suggest small additional modifications prior to publication.
Lines 116, 131, 172, 177, 191, 437: Put “p” in italics. Unify and apply to the entire document.
Table 2, Table 3, Table 4, lines 349, 390, 397, ……: “mL” instead of “ml”. Unify and apply to the entire document.
Lines 307, 382, 395, 419, ….: Include the country of all the companies cited, and cite the companies of all the reagents and equipment’s employed. In case of USA companies, include the city and the state abbreviation including the country. Unify and apply to the entire document.
Lines 318, 346.: Check the format of ºC.
Line 338: In H2O, change the “0” by “O”.
Reviewer 4 Report
The authors corrected the paper in some aspects. However, the manuscript still lacks acceptable phytochemical examination of the analyzed plant material. Presented screening is highly non selective and does not provide reliable data on compounds that were detected. Without improved phytochemical part as requested before, in my opinion the paper should not be published in Plants.